# Investigating Different Grounds Effects on Shock Wave Propagation Resulting from Near-Ground Explosion

**Yan Wang \*, Hua Wang, Cunyan Cui and Beilei Zhao**

Space Engineering University, Beijing 101416, China
\* Correspondence: 08wy@163.com; Tel.: +86-010-66365330

**Abstract:** A massive explosion of a liquid-propellant rocket in the course of an accident can lead to a truly catastrophic event, which would threaten the safety of personnel and facilities around the launch site. In order to study the propagation of near-ground shock wave and quantify the enhancement effect on the overpressure, models with different grounds have been established based on an explicit nonlinear dynamic ANSYS/LS-DYNA 970 program. Results show that the existence of the ground will change the propagation law and conform to the reflection law of the shock wave. Rigid ground absorbs no energy and reflects all of it, while concrete ground absorbs and reflects some of the energy, respectively. Ground may influence the pressure-time curve of the shock wave. When the gauge is close to the explosive, the pressure-time curve presents a bimodal feature, while when the gauge reaches a certain distance to the explosive, it presents a single-peak feature. For gauges at different heights, different grounds may have different effects on the peak overpressure. For gauges of height not greater than 4 m, the impact on the shock wave is obvious when the radial to the explosive is small. On the contrary, as for the gauges of height greater than 4 m, the impact on the shock wave is obvious when the radial to the explosive is big. Ground has the enhancement effect on peak overpressure, but different grounds have different ways. For rigid ground, the peak overpressure factor is about 2. However, for the concrete and soil ground, peak overpressure factor is from 1.43 to 2.1.

**Keywords:** near-ground explosion; shock wave; peak overpressure; LS-DYNA; peak overpressure factor

## 1. Introduction

The launch vehicle is a complex and large-scale technology-intensive system and it plays an important role in the deep space exploration mission. Once an explosion occurs, it would threaten the safety of personnel and facilities around the launch site. Owing to its very low boiling point, liquid propellant is extremely hazardous and its leakage/sudden release from a pressurized tank would lead to vapor cloud formation resulting in subsequent fire and explosion. This point was amply demonstrated by a series of explosion accidents over decades [1]. The study of the rocket shock environment is an ongoing effort to characterize the environment resulting from catastrophic rocket explosion. The purpose is to develop the data and information required to allow launch vehicle designers to develop safer launch systems.

Many workers have studied liquid propellant explosion characteristics by conducting experiments and analyzing experimental data [2–9]. The liquid propellant explosion test has high risk and high cost, and the experimental conditions are strict and the repeatability is poor, which makes it difficult. With the development of computer simulation technology, numerical simulation has become the main means to solve such problems [10–17].

Explosion hazard modeling is a challenging problem since it involves several physical, often complex, interacting aspects. In combustion explosions, the available quantity of energy changes rapidly and is constantly redistributed among heat, the kinetic and chemical energy forms. The problem becomes even more complex when the subsequent shock wave propagation is assumed to interact with the ground, which has a more destructive effect on the personnel and facilities at the launch site. Ground and shock waves interact closely since the shock wave impulses affect the ground, while the ground in turn modifies the shock wave's propagation patterns [18].

Therefore, it is of great significance to investigate different grounds effects on shock wave propagation resulting from near-ground explosion. It is difficult to deduce and calculate the shock wave parameters in theory, so most of the current research adopts the empirical formula based on a large number of tests for prediction. However, due to the limitation of experimental conditions, the results calculated by different empirical formulas often differ greatly, which will bring hidden dangers to the safety analysis of the launch site [19]. However, the explosion test has the characteristics of short time, poor repeatability, and high cost, so it is not convenient for research. Therefore, research on the propagation law of shock wave in complex environment has been investigated by many researchers [20–28].

When a rocket explodes on the launch pad, the shock wave does not travel through the infinite air. The ground has certain influence on the propagation of shock wave. The effect of the ground on the shock wave is reflected in two aspects. On the one hand, the ground will reflect and enhance the shock wave. On the other hand, the earth may absorb some energy of the shock wave [29].

Common types of launching site ground include concrete ground and soil ground. Concrete ground reflects more energy of the shock wave, while soil absorbs more energy of the shock wave and reflects less energy due to its looseness. In order to study the influence of different ground properties on the propagation law of shock wave, this paper considers two extreme cases, one is air material, the other is rigid ground. The air material simulates the shock wave propagation in free space without any obstruction. Rigid ground, on the other hand, absorbs no energy and reflects it all.

In this paper, in order to simulate the propagation of near-ground shock wave and quantify the enhancement effect on peak overpressure, an explicit nonlinear dynamic ANSYS/LS-DYNA Version 970 (Livermore Software Technology Corporation, Livermore, CA, America) program is employed to setup near-ground explosion models with different grounds. The enhancement effect on peak overpressures of different grounds was carried out, which will provide a reliable reference for design and safety evaluation of the launch site.

## 2. Definition of Explosive Yield

TNT equivalent model is to convert the explosive materials into equivalent TNT according to the equal energy, and then the explosive law of TNT is applied to predict the effect of liquid propellant explosion [30]. It is widely used in the field of explosive characteristics of liquid propellant [31]. Given that the mass of liquid propellant $M_0$ and explosive yield $Y$, the TNT $M_T$ can be calculated by Equation (1).

$$M_T = Y \cdot M_0 \tag{1}$$

where, $M_T$ represents mass equivalent to TNT, kg. $Y$ is explosive yield, dimensionless. $M_0$ is mass of liquid propellant, kg.

The object of this paper is a two-stage rocket with four boosters, whose propellant quality of each stage is shown in Table 1. It is very difficult to obtain an accurate explosive yield theoretically under different explosion modes, because the chemical reaction mechanism of liquid propellant is different from that of solid explosive, and is affected by various factors such as the type, mass, detonation time of the propellant, falling speed, and ground property, etc. Through statistical analysis of test data [2–4], the explosive yield of liquid propellant is estimated. This method has certain authority and has been widely used abroad. Explosive yield estimation is mainly calculated by two methods, that is,

the table-lookup-method and chart-reading-method. Therefore, explosive yield under the two methods is calculated respectively and it is finally determined according to the minimum principle [32].

**Table 1.** Equivalent yields at different stages of a certain rocket.

| Items | | Boosters | | Stage-1 | | Stage-2 | |
|---|---|---|---|---|---|---|---|
| | | LOX | RP-1 | LOX | LH$_2$ | LOX | LH$_2$ |
| Propellant mass (kg) | | $4 \times 104{,}000$ | $4 \times 40{,}000$ | 133,300 | 24,360 | 22,200 | 3800 |
| Total mass (kg) | | 576,000 | | 157,660 | | 26,000 | |
| | Table-lookup-method | 0.1 | | 0.6 | | 0.6 | |
| Y | Chart-reading-method | $0.0576 \times 125\% = 0.072$ | | $0.0644 \times 370\% = 0.238$ | | $0.0713 \times 370\% = 0.264$ | |
| | Chosen | 0.072 | | 0.238 | | 0.264 | |

The table-lookup-method is to obtain the explosive yield by reading Table 2, on the basis of determining the propellant type and the explosion mode. While as for the chart-reading-method, explosive power can be obtained from Figure 1 according to propellant mass, and the final explosive yield is obtained by multiplying the specific coefficients corresponding to different propellants in Table 2. The explosive yields are shown in Table 1.

**Table 2.** Explosive yields and specific coefficients of different liquid propellant combinations under different explosion modes.

| Propellant Combination | Explosion Mode | | Mix Proportion | Specific Coefficient |
|---|---|---|---|---|
| | On Launch Pad | Outside Launch Pad | | |
| LOX/LH$_2$ | 60% | 60% | 1:5 | 370% |
| LOX/RP-1 | Within 226.7995t is 20% plus 10% for rest part | 10% | 1:1.25 | 125% |
| N$_2$O$_4$/UDMH | 10% | 5% | 1:2 | 240% |

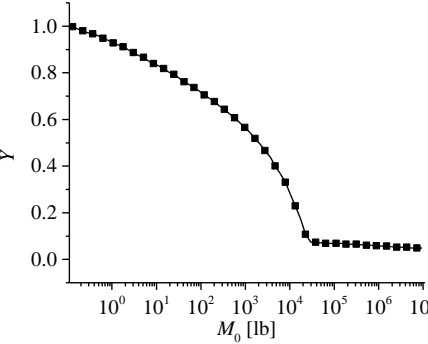

**Figure 1.** The curve between the explosive yields and the mass of propellant.

Combined explosive yields and combined explosive center coordinates are acquired by Equation (2) and (3). The height of the combined explosive center is 21.968 m. Given the height of the launch pad, the height of the true explosive center is 30.668 m.

$$y_{zh} = \sum m_i y_i / \sum m_i \qquad (2)$$

where, $y_{zh}$ is the combined explosive yield, $y_i$ is the explosive yield of stage $i$, $m_i$ is total propellant mass of stage $i$, whose unit is kg.

$$x_z = \sum m_i x_i / \sum m_i \qquad (3)$$

where, $x_i$ is the distance between the middle of tanks of stage $i$ and the explosive center, whose unit is m.

## 3. Finite Element Model of Near-Ground Explosion

### 3.1. Governing Equations and Computation Code

In the continuum mechanics, the momentum equation is

$$\sigma_{ij,j} + \rho f_i = \rho \ddot{x}_i \tag{4}$$

where, $\sigma_{ij,j}$ is one order partial derivative of Cauchy stress component $\sigma_{ij}$ with respect to the coordinate variable in the $j$th direction, $f_i$ is the volume force vector of the unit mass, $\ddot{x}_i$ is the acceleration vector, and $\rho$ is the density of air.

The conversation equation of mass is

$$\rho = J\rho_0 \tag{5}$$

where, $\rho_0$ is the initial density of air, $J = \left| \frac{\partial xi}{\partial xj} \right|$, $x_i$ is the space coordinate in the $i$th direction, $x_j$ is the matter coordinate in the $j$th direction, and $\partial x_i / \partial x_j$ is the strain gradient with respect to $x_j$.

The energy equation is

$$\dot{E} = VSij\dot{\varepsilon}ij - (p + q)\dot{V} \tag{6}$$

where, $V$ is volume, $\dot{\varepsilon}ij$ is strain rate tensor, and $q$ is artificial viscidity. Deviatoric stress tensor is given by

$$Sij = \sigma ij + (p + q)\varepsilon ij \tag{7}$$

where, $\varepsilon_{ij}$ is Kronecher symbol and $p$ is hydrodynamic pressure, given by

$$p = -\frac{1}{3}\sigma kk - q. \tag{8}$$

ANSYS/LS-DYNA 970 is a finite element computation code for dynamics suitable for impact, explosion, and other instantaneous problems, and is used to carry out the numerical simulation process. In LS-DYNA, there are three classes of methods available for generating the grids and mesh, including Eulerian, Lagrangian, and arbitrary Lagrangian–Eulerian (ALE) method. In this study, the ALE method is employed to simulate the process of the TNT explosion, the propagation of air shock waves, and the interaction of explosion shock waves with the grounds. The ALE algorithm consists of a classical Lagrangian step in which the mesh moves along with the modeled material, a rezone step in which the mesh is modified to preserve good quality through the computation of the Eulerain time step, and a remapping step in which the solution is conservatively transferred from the old mesh to the new rezoned one, and thereby performs the fluid–solid coupling transient analysis [33].

### 3.2. Modeling Geometry and Meshing

In order to investigate the impact of different grounds on the shock wave propagation law when the rocket exploded on the launch pad, the whole rocket is selected to carry out numerical simulation. Numerical models of four different materials are chosen, including concrete ground, soil ground, rigid ground, and air. Model 1 to Model 3 are the TNT that explodes above the rigid ground, the concrete ground and the soil ground, respectively, while Model 4 is the TNT that explodes in the air. In Model 1, air domain is a Φ 258 m × 40 m cylinder and the ground is rigid constraint. In Model 2 and Model 3, air domain which is a Φ 258 m × 50 m cylinder and ground domain which is a Φ 258 m × 10 m cylinder are established. Model 4 only includes air domain which is a Φ 258 m × 50 m cylinder. After considering the symmetry of the simulation model, a 1/4 symmetrical geometrical model is adopted. In order to reduce the calculation amount, the air domain is divided into three parts according to the radial, which are 0~30 m, 30~70 m, 70~129 m. The mesh size of the air domain is 0.58 m, 1.16 m, and 2.32 m, respectively. The TNT and the ground are divided into mesh sizes of 0.20 m and 2.32 m.

The height of the detonation center to the ground surface is 30.668 m. The finite element model and gauges distribution are shown in Figure 2.

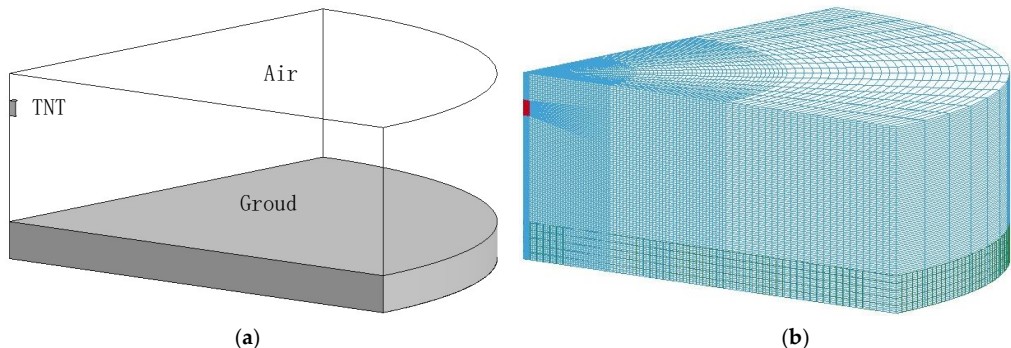

(**a**)　　　　　　　　　　　　　　　　　　　　(**b**)

**Figure 2.** Physical model and finite element model of near-ground explosion. (**a**) Physical model; (**b**) Finite element model.

It is assumed that the radius of the TNT cylinder is $r_0$ (m) and the height is $h_0$ (m). Therefore, the length-diameter ratio of TNT is $h_0/2r_0$. Given the mass of TNT, $r_0$, and $h_0$ of TNT cylinder can be calculated by using Equation (10) which is derived from Equation (9). Unit system of cm-g-ms is adopted in the numerical model. The explosive source is located at the center of mass and a point-detonation-method is used [34].

$$\begin{cases} M_T = \rho \cdot V \\ V = \pi r_0^2 h_0 \\ h_0/2r_0 = 1 \end{cases} \tag{9}$$

$$r = \sqrt[3]{\frac{M_T}{2\pi\rho}} \tag{10}$$

Eight-node element of Solid 164 is adopted for the 3D explicit analysis. In order to prevent the element distortion in large deformation and nonlinear structural analyses, ALE algorism is used. TNT and air are modeled with ALE multi-material grids, but different grounds with Lagrangian grids. Fluid–solid coupling algorithm is used between two grids [35].

Furthermore, the transitional displacement of the nodes normal to the symmetry planes is constrained, in order to simulate the propagation effect of shock wave in the symmetric plane. Non-reflecting boundary condition is applied to top and lateral surfaces, which allows the fluid medium to flow out in order to simulate the effect of infinite space [36]. The displacement constraint is applied in the *z* direction of the ground, while rigid constraint is applied on the rigid ground.

As are shown in Figure 3, gauges in the model are set at five different heights, which are from 2 m to 10 m with an interval of 2 m. Twelve gauges are set at each height, whose *x* is from 0 m to 110 m, with an interval of 10 m.

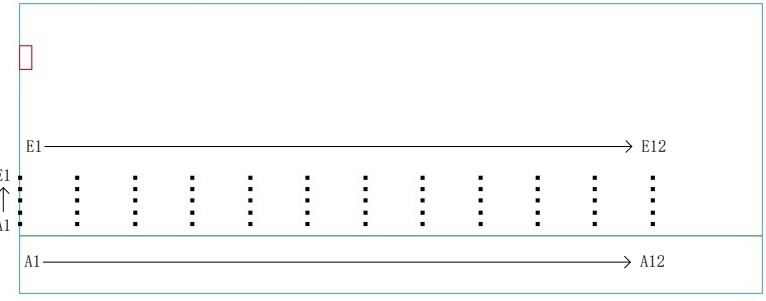

**Figure 3.** Distribution of gauges at different heights.

### 3.3. Material Models and Equation of State

Air is commonly modeled by *MAT_NULL with a linear polynomial equation of state, EOS_LNIEAR_POLYNOMIAL. *MAT_NULL is material type 9 in LS-DYNA 970 code. This material allows equations of state to be considered without computing deviatoric stresses. Optionally, a viscosity can be defined. Also, erosion in tension and compression is possible. Air and water usually use this model. This model avoids the calculation of stress and strain. EOS_LNIEAR_POLYNOMIA defines the pressure by

$$P = C_0 + C_1\mu + C_2\mu^2 + C_3\mu^3 + \left(C_4 + C_5\mu + C_6\mu^2\right)E_0 \tag{11}$$

where, the parameter $\mu$ is defined as $\mu = \rho/\rho_0$, $\rho$ is the current density, and $\rho_0$ is a nominal or reference density; $C_0$–$C_6$ are the equation coefficients; and the parameter $E_0$ is the initial internal energy of reference specific volume per unit. The parameters used are shown in Table 3 [37,38].

**Table 3.** Parameters of the air.

| $\rho$ (g/cm$^3$) | $C_0$ | $C_1$ | $C_2$ | $C_3$ | $C_4$ | $C_5$ | $C_6$ | $E_0$ (J/m$^3$) | $V_0$ |
|---|---|---|---|---|---|---|---|---|---|
| 1.29 | $-1.0 \times 10^{-6}$ | 0 | 0 | 0 | 0.4 | 0.4 | 0 | 25 | 1.0 |

TNT is modeled by the high explosive material model, MAT-HIGH-EXPLOSLVE-BURN, it allows the modeling of the detonation of a high explosive. In addition, an equation of state must be defined. Burn fractions, *F*, which multiply the equations of states for high explosives, control the release of chemical energy for simulating detonations. The Jones–Wilkins–Lee (JWL) equation of state defines pressure by

$$P = A\left(1 - \frac{\omega}{R_1 V}\right)e^{-R_1 V} + B\left(1 - \frac{\omega}{R_2 V}\right)e^{-R_2 V} + \frac{\omega E}{V} \tag{12}$$

where, $A$, $B$, $R_1$, $R_2$, $\omega$ are the equation coefficients and $V$ is the initial relative volume.

Parameters selected are shown in Table 4 [39].

**Table 4.** Parameters of TNT charge.

| $\rho$ (g/cm$^3$) | $v_{\mathrm{D}}$ (km/s) | $P_{\mathrm{CJ}}$ (GPa) | $A$ (GPa) | $B$ (GPa) | $R_1$ | $R_2$ | $\omega$ | $E_0$ (J/m$^3$) | $V_0$ |
|---|---|---|---|---|---|---|---|---|---|
| 1.63 | 6.93 | 27 | 371 | 7.43 | 4.15 | 0.95 | 0.3 | $7 \times 10^3$ | 1.0 |

The concrete ground is modeled using MAT_PLASTIC_KINEMATIC [40]. This model can describe the isotropic hardening and follow-up hardening plastic models, and can also consider the influence of strain rate. It is applicable to beam, shell, and solid units, and the calculation efficiency is very high. In the plastic kinematic material model, the mechanical properties are characterized by the material yield strength ($\sigma_0$), the Young's modulus ($E$), and the tangent modulus ($E_t$: The slope of the stress–strain curve in the plastic region). The yielding is defined by von Mises yield criterion. In the plastic kinematic plasticity algorithm, the flow stress ($\sigma$) is given as

$$\sigma_y = \left[1 + \left(\frac{\dot{\varepsilon}}{C}\right)^{\frac{1}{P}}\right]\left(\sigma_0 + \beta E_P \varepsilon_p^{eff}\right) \tag{13}$$

where, $\dot{\varepsilon}$ is the strain rate, $C$ and $P$ are the strain rate parameters, $\beta$ is the hardening coefficient and $\varepsilon_p^{eff}$ and $E_P$ are the effective plastic strain and plastic hardening modulus, respectively. The plastic hardening modulus is calculated by using the following relation

$$E_P = \frac{EE_t}{E - E_t} \tag{14}$$

The parameters of the concrete used are shown in Table 5.

**Table 5.** Parameters of the concrete [41].

| $\rho$/(g/cm$^3$) | $E$/GPa | $\mu$ | $\sigma_0$/GPa | $E_P$/GPa | $\beta$ | $C$ | $P$ | FS | VP |
|---|---|---|---|---|---|---|---|---|---|
| 2.65 | 4.0e × 10$^{-1}$ | 0.3 | 1.0e × 10$^{-3}$ | 4.0e × 10$^0$ | 0.50 | 0 | 0 | 0.8 | 0.8 |

The soil ground is modeled using MAT_SOIL_AND_FOAM [42]. Due to its simplicity and added flexibility, this model has a shear failure surface that is pressure dependent, which is a basic property of geo-material. It is more fluid-like under many conditions, which is an idea for soft soil. The pressure dependent shear strength envelope is written in terms of a quadratic in pressure by the equation:

$$\phi = J_2 - (a_0 + a_1 p + a_2 p) \tag{15}$$

where, $J_2 = S_{ij}S_{ji}/2$, $S_{ij}$ and $S_{ji}$ are the stress tensors, $p$ is the mean pressure, $a_0$, $a_1$, and $a_2$ are constants which are determined from the triaxial compression tests. The parameters of the soil are shown in Table 6.

**Table 6.** Parameters of the soil [43].

| $\rho$ (g/cm$^3$) | $G$/GPa | $B$ | $a_0$/GPa | $a_1$/GPa | $a_2$/GPa | $\varepsilon^P_{e2}$ | $\varepsilon^P_{e3}$ |
|---|---|---|---|---|---|---|---|
| 1.8 | 1.60 × 10$^{-6}$ | 1.382 | 3.3 × 10$^{-5}$ | 1.31 × 10$^{-9}$ | 1.23 × 10$^{-3}$ | 0.05 × 10$^{-2}$ | 9 × 10$^{-4}$ |
| $\varepsilon^P_{e4}$ | $\varepsilon^P_{e5}$ | $\varepsilon^P_{e6}$ | $\varepsilon^P_{e7}$ | $\varepsilon^P_{e8}$ | $\varepsilon^P_{e9}$ | $\varepsilon^P_{e10}$ | $P_2$/GPa |
| 1.1 × 10$^{-3}$ | 1.5 × 10$^{-3}$ | 1.9 × 10$^{-3}$ | 2.1 × 10$^{-3}$ | 2.2 × 10$^{-3}$ | 2.5 × 10$^{-3}$ | 3.0 × 10$^{-3}$ | 3.42 × 10$^{-4}$ |
| $P_3$/GPa | $P_4$/GPa | $P_5$/GPa | $P_6$/GPa | $P_7$/GPa | $P_8$/GPa | $P_9$/GPa | $P_{10}$/GPa |
| 4.53 × 10$^{-4}$ | 6.76 × 10$^{-4}$ | 1.27 × 10$^{-4}$ | 2.08 × 10$^{-4}$ | 2.71 × 10$^{-4}$ | 3.92 × 10$^{-4}$ | 5.66 × 10$^{-4}$ | 1.23 × 10$^{-4}$ |

## 4. Propagation Law of Near-Ground Explosion Shock Wave

### 4.1. Verification of Blast Model

Figure 4 shows the empirical data and numerical data about the peak overpressure along *x* at the height of 2 m. As can be seen, the prediction values of each empirical formula in the near-field of the explosion are quite different, which is mainly because the detonation products in the near-field are quite different from the test values. In the far-field of the explosion, the influence of detonation products decreases and the predicted values of each empirical formula tend to be consistent. When scaled distance is small, different empirical formulas have a large deviation because of the test error. While when the scaled distance is big, different empirical formulas have a large consistency [44–46].

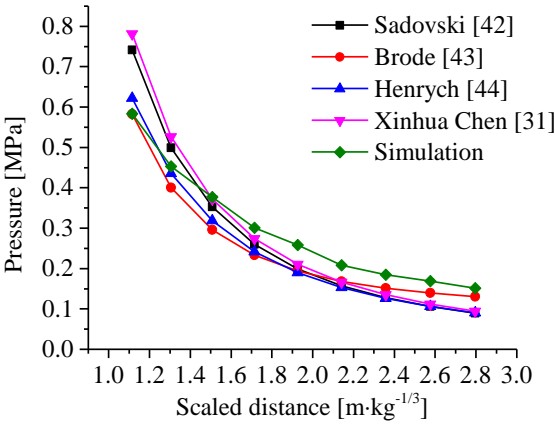

**Figure 4.** Comparison of theoretical and numerical data in air explosion at the height of 2 m.

By comparison, simulation data is almost the same as the empirical data, and the deviation is within the permissible scope of the project. Certain deviation between the simulation result and the empirical formula is mainly because that the empirical formula is obtained by fitting according to the test results, while the simulation model is a relatively ideal result obtained by the calculation of the aerodynamic equation, and there are certain differences between the test conditions and the simulation models.

### 4.2. Reflection of Shock Wave above Ground

When the explosive detonates at the height *h* above the ground, the air shock wave generated spreads in all directions in a spherical shape. When the shock wave reaches the ground, it will be reflected above it

When the incident angle is approximately equal to zero, the front of the incident shock wave is parallel to the ground, so a positive reflection occurs. When the incident is greater than zero, an oblique reflection of the shock wave occurs. When the incident angle continues to increase to the limit, the Mach reflection occurs, and the incident wave and the reflected wave become Mach waves. Reference [47] presents the functional relationship between the limit incidence angle and the reciprocal of the proportional height measured by the test. According to law, the limit incidence angle of the models in this paper is about 40°.

### 4.3. Propagation Law of the Shock Wave above Rigid Ground

Visualization of pressure contours available during the post-processing stage allows for a better understanding of complex process of shock wave interaction with the ground. Figure 5 shows the pressure contours at different moments, which reappears in the interaction process between the shock wave and the rigid ground [48]. Different images represent typical propagation moments of shock wave, and different pressure values are represented by different colors. A strong chemical reaction takes place after detonation, resulting in a sharp expansion and diffusion in all directions of the high temperature and high pressure explosive products, and shock wave propagates outward in the form of a spherical wave. As shown in Figure 5a, the air shock wave encounters the rigid ground and a local high pressure zone near the ground is formed. '*x*' is defined as the radial from the gauge to the explosive center. When *x* is equal to zero, the incident angle of the shock wave is equal to zero, and a positive reflection will occur. This is because the air particle at the shock wave front is hindered and its velocity is reduced, which is superimposed with the later wave to form the enhanced wave near the ground. As shown in Figure 5b, the shock wave overlaps near the ground and a high pressure zone is formed. This is because when *x* is less than 44.116 m, the incident angle is less than the limit incident angle of 40°, and regular reflection occurs. As shown in Figure 5c, the area of high pressure gradually begins to rise off the ground. This is because when *x* is greater than 44.116 m, the shock wave incident angle is greater than the limit incident angle of 40°, and Mach reflection will occur. As shown in Figure 5d, the height of the Mach bar gradually increases. This is because the pressure of the shock wave decreases when *x* increases, and gradually approaches to the plane wave.

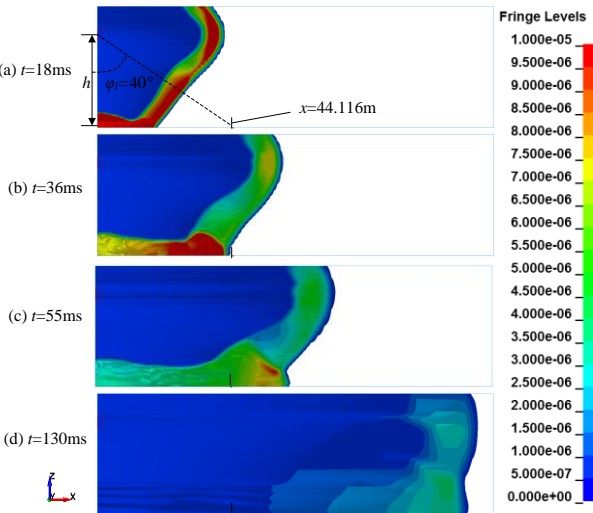

**Figure 5.** Reflected pressure contours images of the shock wave explosion above the rigid ground.

## 5. Results and Analysis

### 5.1. The Effect on the Propagation of Near-Ground Shock Wave

As shown in Figure 6, the pressure contour images of the shock wave above the concrete ground are given. When $t = 120$ ms, the air shock wave will be stratified when it encounters the ground. Some energy of the shock wave will be transmitted into the ground, and some will be reflected, superimposing with the incident wave near the ground. When $t = 255$ ms, the shock wave under the ground precedes that in the air, this is because the shock wave travels much faster in the solid than in the gas. When $t = 675$ ms, a high pressure area appears near the ground due to Mach reflection. When $t = 1140$ ms, the height of the Mach bar gradually rises, and the pressure near the ground is higher than other areas in the air. It indicates that the ground has an enhanced effect on the propagation of the shock wave, and the enhanced area is mainly concentrated near the ground. In the face of the enhancement effect of the shock wave above different grounds, it is necessary to conduct in-depth research through data analysis.

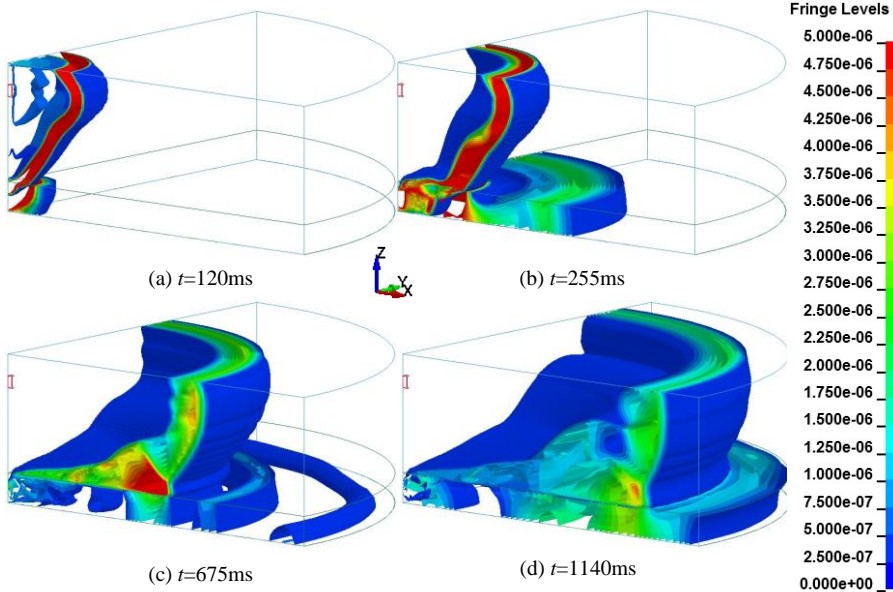

**Figure 6.** Pressure contours images of the shock wave above the concrete ground.

*5.2. The Effects on the Pressure-Time Curves of Shock Wave above Different Grounds*

Figures 7–10 respectively show the pressure-time curves of the shock wave of gauges A1 to A12 when the explosion occurs above different grounds. As shown in Figure 7, shock wave pressure of different gauges rises linearly and then decreases gradually with time. The pressure curve shows the single-peak characteristic, and the peak overpressure decreases gradually with the increase of *x*. This is mainly because the scaled distance to explosion center increases from A1 to A12. Different from the explosion pressure curves explosion in the air, the gauges in Figures 8–10 are divided into two groups according to the characteristics of the pressure-time curves. Group 1 is the gauges from A1 to A4, while Group 2 is the gauges from A5 to A12. The pressure-time curves of the two groups show different characteristics. The pressure-time curves in Group 1 have obvious bimodal characteristics. The first peak is the incident wave, and the second one is the reflected wave. As is shown in Figure 8, the smaller the *x* is, the higher reflected peak overpressure is. For gauges A1 and A2, the peak pressure of the reflected shock wave is larger than that of the incident one, while the result is opposite for gauges A3 and A4. This is mainly because the smaller *x* is, the smaller the incident angle is, and the higher the energy of the reflected shock wave will be. The pressure-time curves in Group 2 show a single peak characteristic. This is because the Mach reflection is formed at this zone, and the peak overpressure is the maximum pressure of the composite wave. As can be seen, the peak pressure of the gauge at the same position is higher than that in the air. As is shown in Figure 9, for gauges A1 and A2, the bimodal characteristics of the pressure-time curves are not obvious. While for gauges A3 and A4, and the pressure-time curves, have obvious bimodal characteristics. The peak pressure of incident wave and reflected wave is relatively close, but obviously lower than the peak pressure of rigid ground. This is mainly because when the incident angle is small, more energy is sent into the ground, so the reflected wave is not obvious. However, with the increase of the incident angle, the energy sent into the ground by the shock wave decreases, so the reflected wave is obvious. The pressure-time curves in Group 2 show a single peak characteristic like the curves in Figure 8b, but there is just a decrease in the overpressure at the gauge. The curve in Figure 10 shows a similar pattern to that in Figure 9, except that there are some differences in the pressure values.

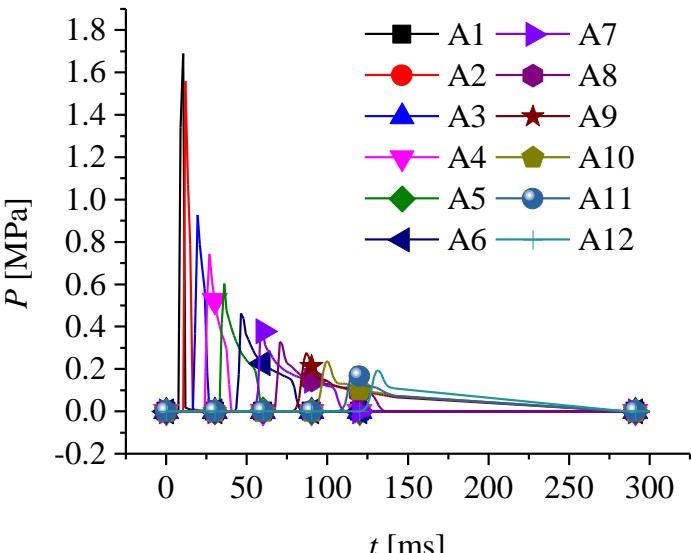

**Figure 7.** Pressure-time curves at 2 m gauges of shock wave in the air.

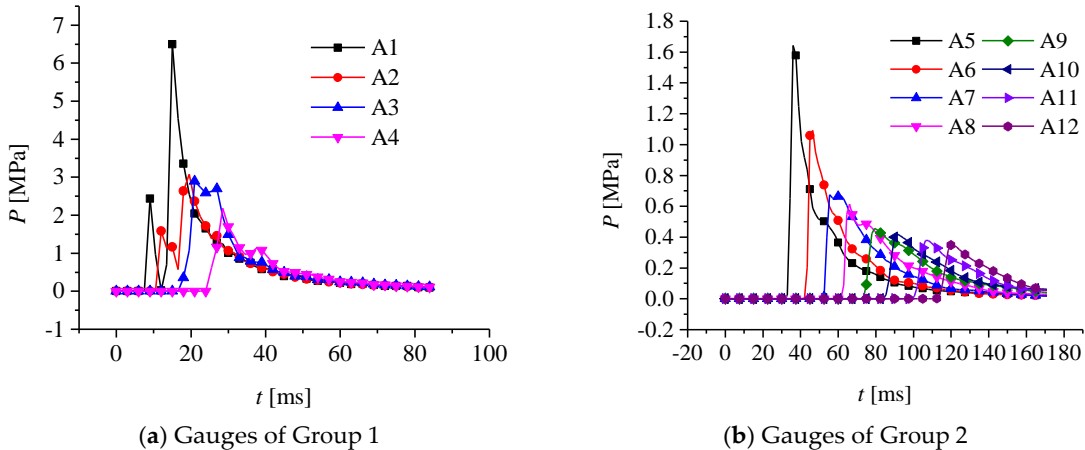

**Figure 8.** Pressure-time curves at 2 m gauges of shockwave above the rigid ground.

(**a**) Gauges of Group 1

(**b**) Gauges of Group 2

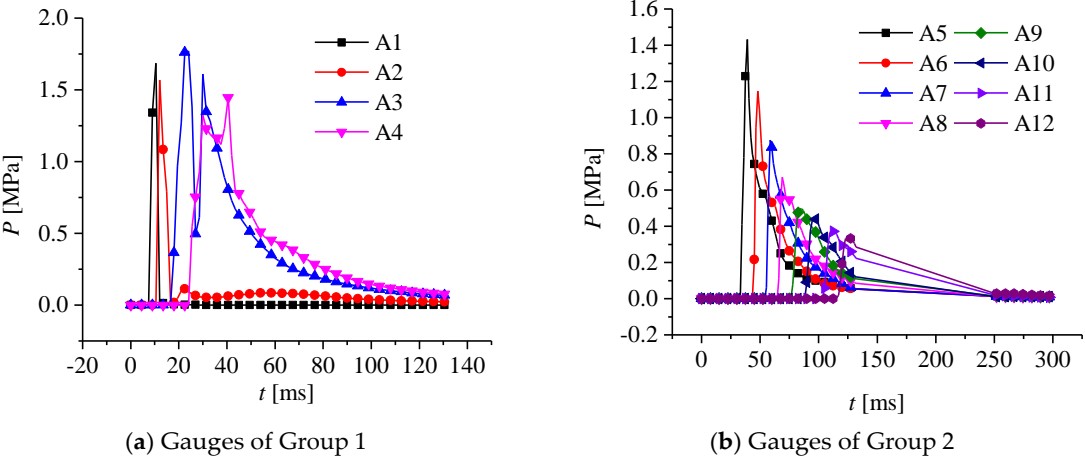

(**a**) Gauges of Group 1

(**b**) Gauges of Group 2

**Figure 9.** Pressure time curves at 2 m gauges of shock wave above the concrete ground.

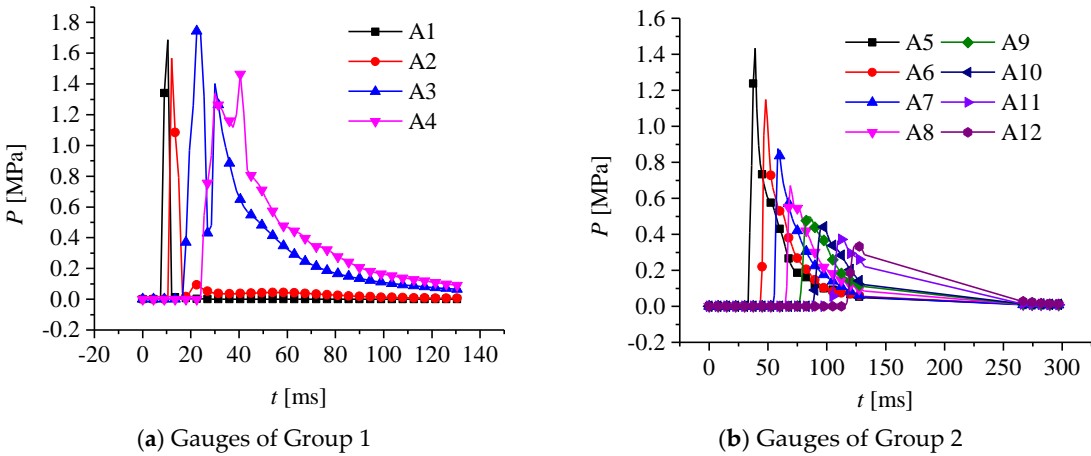

(**a**) Gauges of Group 1

(**b**) Gauges of Group 2

**Figure 10.** Pressure-time curves at 2 m gauges of shock wave above the soil ground.

It can be seen from Figures 8–10 that, the pressure curves of gauges from A1 to A4 have two peaks. The first one is the peak pressure of the incident wave, and the second one is the peak pressure of the shock wave reflected by the ground. However, for gauges from A5 to A12, this phenomenon is not obvious. Only one obvious peak pressure appears, indicating that Mach reflection occurs. Therefore, for the shock wave generated by the explosion above the ground, Mach reflection starts to occur at the incident angle of about 40°, which has nothing to do with the property of the ground.

### 5.3. The Effects on the Peak Overpressure of Shock Wave above Different Grounds

As shown in Figure 11a, the peak overpressure curves of gauges at 2 m height for different grounds present different rules with $x$. For rigid ground, when $x$ is not greater than 50 m, the peak overpressure of the shock wave is larger than that of others. However, when $x$ is greater than 50 m, the peak overpressure gradually approaches that of the concrete ground and soil ground. This is due to the fact that the rigid ground does not absorb energy of the shock wave, but reflects the shock wave and generates regular and irregular reflections with the incident waves, enhancing the shock wave. The enhancement effect is more obvious when $x$ is smaller, and less when $x$ increases. For the explosion above concrete ground and soil ground, when $x$ is not greater than 10 m, the peak overpressure of shock wave is very close to that in the air. However, when $x$ is greater than 10 m and less than 40 m, peak overpressure starts to be higher than that in the air, and gradually approaches to the peak overpressure of rigid ground as $x$ increases. When $x$ is greater than 40 m, the peak overpressures of the shock wave above different grounds gradually approach, but all of them are higher than the peak overpressure of explosion in the air.

Figure 11b presents the peak overpressure curves of gauges at 4 m height. For rigid ground, when $x$ is not greater than 70 m, the peak overpressure is higher than that of others. However, when $x$ is greater than 70 m, the peak overpressure is gradually close to that of concrete and soil ground. For concrete and soil ground, when $x$ is not greater than 20 m, the peak overpressure of shock wave is very close to that in the air. However, when $x$ is greater than 20 m and less than 60 m, peak overpressure starts to be higher than that in the air, and gradually approaches to that of rigid ground. When $x$ is greater than 60 m, the peak overpressures of the shock wave above different grounds gradually approach.

Figure 11c presents the peak overpressure curves of gauges at 6 m height. For rigid ground, when $x$ is not greater than 20 m, the peak overpressure of the shock wave above different grounds tends to be consistent. However, when $x$ is greater than 20 m and not greater than 100 m, the peak overpressure is higher than that of concrete ground and soil ground. When $x$ is greater than 100 m, the peak overpressure is gradually close to that of concrete ground and soil ground. For concrete ground and soil ground, when $x$ is not greater than 30 m, the peak overpressure of the shock wave is very close to that in the air. However, when $x$ is more than 30 m, peak overpressure starts to be higher than that in the air, and gradually approaches to the peak overpressure of rigid ground as $x$ increases.

Figure 11d presents the peak overpressure curves of gauges at 8 m height. For rigid ground, when $x$ is not greater than 30 m, the peak overpressure of the shock wave above different grounds tends to be consistent. However, when $x$ is greater than 30 m, the peak overpressure is higher than that of concrete ground and soil ground. For concrete ground and soil ground, when $x$ is not greater than 30 m, the peak overpressure of the shock wave is very close to that in the air. However, when $x$ is greater than 30 m, peak overpressure starts to be higher than that in the air, but is lower than that above the rigid ground. As for Figure 11e, the law of peak overpressure curves is generally similar to that in Figure 11d, except for the value of the peak overpressure.

From what has been discussed above, when $h$ is not greater than 4 m, the peak overpressure generated by the explosion above the rigid ground is higher when $x$ is relatively small, and the enhancement effect on the overpressure is no longer obvious with the increase of $x$. When $x$ is small, the peak overpressure of concrete ground and soil ground is the same as that in the air. However, when $x$ increases to a certain extent (10 m or 20 m), the enhancement effect on the shock wave gradually becomes prominent, which is consistent with that of rigid ground. As for $h$ greater than 4m and less than 10 m, the peak overpressure curves of different grounds are the same when $x$ is small, while the differentiation begins when $x$ reaches a certain value. The peak overpressure of rigid ground is higher than that of concrete and soil ground, and the peak overpressure of concrete and soil ground is higher than that of the air. This is mainly because when $x$ is relatively small, the incident wave plays the main role, while when $x$ is relatively big, it is the superimposed Mach wave that plays a major role.

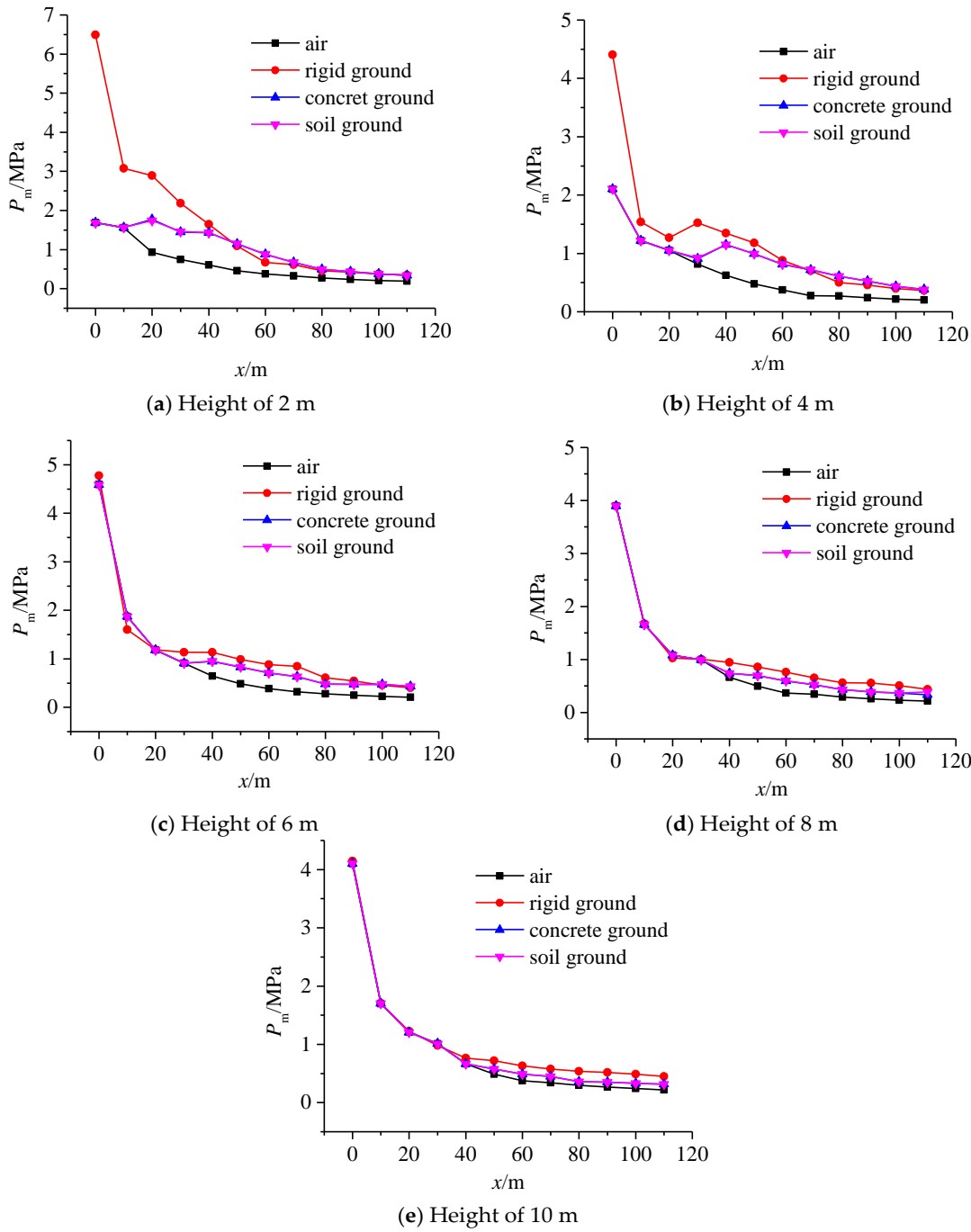

**Figure 11.** Distribution of peak overpressure of gauges at different height produced by explosion on different ground.

## 5.4. The Peak Overpressure Factor of Different Ground

It is assumed that the peak overpressure of gauges in Model 1 to Model 3 is $P_{ground}$, and the peak overpressure of gauge in Model 4 is $P_{air}$, then "peak overpressure factor" is defined in this paper, which is $\alpha = P_{ground}/P_{air}$. It reflects the enhancement effect for peak overpressure of different grounds.

As shown in Figure 12a, for rigid ground, when $x$ is small, $\alpha$ at different heights varies greatly. The lower the height, the greater the $\alpha$ is, and the maximum reaches 3.8. With the increase of $x$, $\alpha$ of different heights gradually approaches, and eventually approaches 2. As shown in Figure 12b,c, for the concrete and soil ground, with the increase of height, $\alpha$ increases successively from 1, the lower the

height, the greater the rise will be, and then starts to stabilize at a specific value. The maximum $\alpha$ with a height of 6 m is 2.1, and the minimum $\alpha$ with a height of 10 m is 1.43.

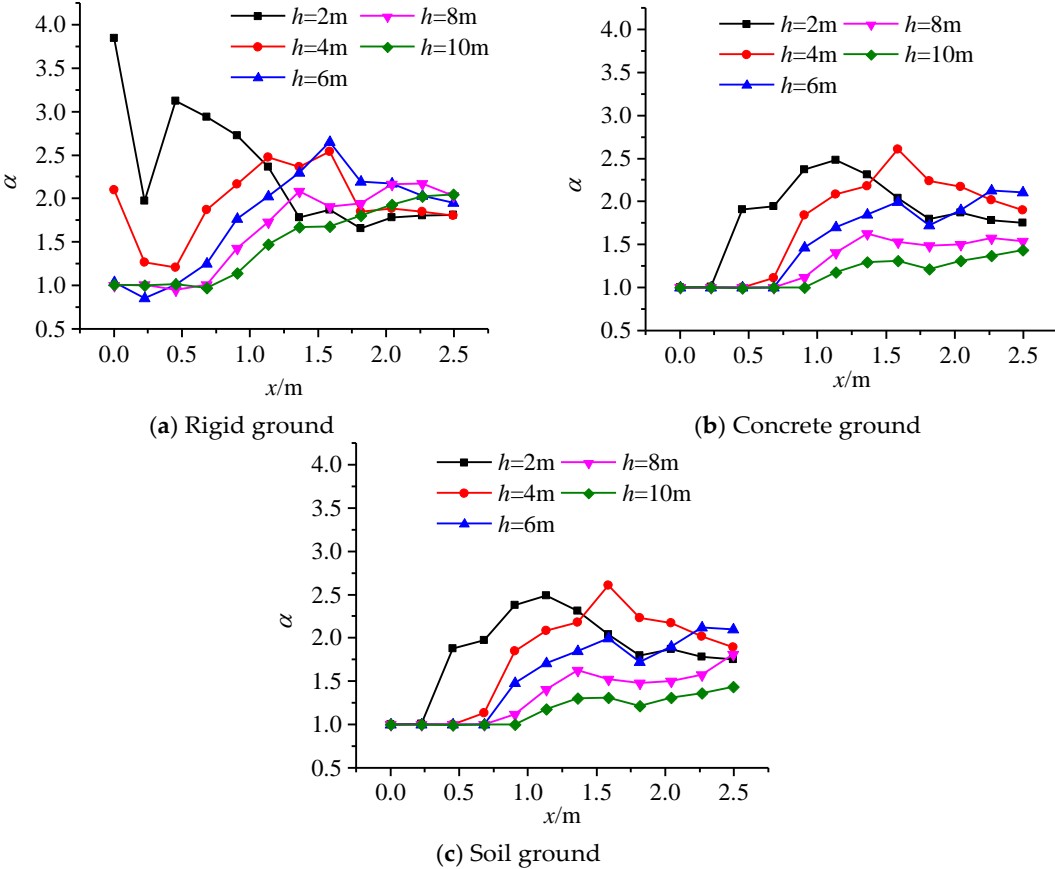

**Figure 12.** The peak overpressure factor of the explosion above different ground.

It can be seen from Figure 12a–c that, for rigid ground, the enhancement effect of peak overpressure with different height is roughly the same, and the peak overpressure factor is about 2. However, the concrete ground and soil ground have different effects on the peak overpressure factors at different heights. Among them, peak overpressure factor of 6m height is the largest, reaching 2.1, while the factor of 10 m height is the smallest, which is 1.43.

## 6. Conclusions

A numerical method for simulating rocket explosion on the launch pad was developed in order to study the propagation law of near-ground explosion and quantify the enhancement effect on peak overpressure. From analysis and discussion of results, the following main conclusions were drawn:

(1)  The existence of the ground will change the propagation law and conform to the reflection law of the shock wave. Rigid ground absorbs no energy and reflects all of it, while concrete ground absorbs and reflects some of energy, respectively. However, the shock waves are all enhanced at the near-ground of different materials.

(2)  Ground may have certain effect on the pressure-time curve of the shock wave. The pressure–time curves of shock wave in the air show a single-peak feature. If the ground exists, the pressure–time curve of the gauge whose incident angle is less than 40° presents a bimodal feature, namely, incident wave and reflected wave. However, when the incident angle of the gauge is greater than 40°, the pressure–time curve shows a single-peak feature, and the peak value is Mach reflected pressure.

(3) For gauges at different heights, different ground may have different effects on the peak overpressure. For gauges whose height is not greater than 4 m, when $x$ is small, the peak overpressure above the rigid ground is the highest. With the increase of $x$, the peak overpressure above the three kinds of ground gradually tends to be the same. As for $h$ greater than 4 m, the peak overpressure curves of different ground are the same when $x$ is small. With the increase of $x$, the peak overpressure of rigid ground is higher than that of concrete and soil ground, and the peak overpressure of concrete and soil ground is higher than that of the air.

(4) Grounds have the enhancement effect on peak overpressures, but different grounds have different ways. For rigid ground, the enhancement effect of peak overpressure at different heights is nearly the same and the peak overpressure factor is about 2. However, for the concrete and soil ground, the enhancement effect of the peak overpressure at different heights-TNT is quite different, among which the peak overpressure factor of 6 m is about 2.1, and the peak overpressure factor of 10m is the smallest, about 1.43.

**Author Contributions:** Methodology, Y.W., H.W.; software, Y.W., B.Z.; formal analysis, C.C.; investigation, Y.W.; resources, Y.W., H.W., C.C.; data curation, Y.W.; writing—original draft preparation, Y.W.; writing—review and editing, Y.W.; visualization, Y.W.

**Funding:** This research received no external funding.

**Acknowledgments:** The authors greatly appreciate the comments from the reviewers, whose comments helped to improve the quality of the paper.

**Conflicts of Interest:** The authors declare no conflict of interest.

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
