# Peer review of "Investigating Different Grounds Effects on Shock Wave Propagation Resulting from Near-Ground Explosion"

_applsci, doi:10.3390/app9173639_

Round 1
Reviewer 1 Report
The paper presents and interesting research on the explosion wave propagation above the ground of different mechanical properties (rigid, concrete, soil). Simulations are performed by well-established software ANSYS/LS-DYNA. The formulation and models used are adequate. Conclusions are supported by the results obtained.
Few points need to be improved, though.
Line 89: the title of abscissa axis of Fig.1 is not understandable The graph in Figure 1 looks strange, it must be drawn by few point, not a straight line with a kink. Line 159: what is "null material"? Lines 194-196: "simulation results show ... large consistency in the far-field" is not true, the difference is observed in the near and far field, as can be seen from Fig. 4 How was the data presented in Fig. 6 obtained? Was it borrowed from literature? A clear explanation must be given. Line 258: why is the pressure behind the shock decreases exponentially? Is this a strict result, or just visual observation? Maybe it is better to say that "the pressure decreases gradually"? Figure 3 introduces notation A1-E12 for the pressure gauges, however, these labels are not used when describing the results (Section 5). Either Figure 3 is redundant, or the labels must be used in the text. The term "multiplication coefficient" must be explained in the text. Perhaps, a better word would be "peak pressure factor" or something similar. The phrase "Grounds have the multiplication effect on peak overpressures" does not sound right at all.Also, authors might improve paper readability by paying more attention to English (line 63 - Definition, not Definite; line 76 - "which has certain authority" is not understandable; etc.)
Author Response
Dear professor:
I am the corresponding author of the manuscript “Investigating different grounds effects on shock wave propagation resulting from near-ground explosion”(ID applsci-565271). We must thank you for the critical comments and constructive recommendations. We feel lucky that our manuscript went to you as the valuable comments from you not only helped us with the improvement of our manuscript, but also raised some thoughtful suggestions.
Based on your comments and suggestions, we have made significant modification on the original manuscript. We have asked for native English speakers to revise the paper before it was submitted this time. We hope the new manuscript will meet your magazine’s standard.
Here are our one-by-one responses to your comments.
Point 1: Line 89: the title of abscissa axis of Fig.1 is not understandable. The graph in Figure 1 looks strange, it must be drawn by few point, not a straight line with a kink.
Response 1: We are very grateful for your insightful comments and questions, and we are sorry for this confusion. Figure 1 shows the relationship between the explosive yield Y and the mass of propellant M0. The abscissa axis represents the total mass of the propellant in pounds, which has a certain conversion relationship with the unit of kilograms (1lb≈0.4536kg). We are so sorry for that the original expression is easy to confuse the readers. The graph and the title of abscissa axis of Figure 1 have been modified in revised version.
Point 2: Line 159: what is "null material"?
Response 2: *MAT_NULL is material type 9 in LS-DYNA 970 code. This material allows equations of state to be considered without computing deviatoric stresses. Optionally, a viscosity can be defined. Also, erosion in tension and compression is possible. Air and water usually use this model.
Point 3: Lines 194-196: "simulation results show ... large consistency in the far-field" is not true, the difference is observed in the near and far field, as can be seen from Fig. 4 How was the data presented in Fig. 6 obtained? Was it borrowed from literature? A clear explanation must be given.
Response 3: Sorry for this confusion. As can be seen from Figure 4, when scaled distance is small, different empirical formulas have a large deviation because of the test error. While when the scaled distance is big, different empirical formulas have a large consistency. This conclusion can be seen in References 42 to 44. Figure 6 refers to the conclusion in reference 45. This curve is obtained by fitting data obtained through a large number of experiments. The deviation between the simulation result and the empirical formula is within the engineering allowable range.
Point 4: Line 258: why is the pressure behind the shock decreases exponentially? Is this a strict result, or just visual observation? Maybe it is better to say that "the pressure decreases gradually"?
Response 4: Thank you very much for your insightful and valuable comments. The phrase "exponentially" is replaced by the "gradually".
Point 5: Figure 3 introduces notation A1-E12 for the pressure gauges, however, these labels are not used when describing the results (Section 5). Either Figure 3 is redundant, or the labels must be used in the text.
Response 5: Sorry for this confusion. The purpose of Figure 3 is to show the distribution of the pressure gauges set up in the simulation. Gauges in the model are set at five different heights, which are from 2m to 10m with an interval of 2m. Twelve gauges are set at each height, whose x is from 0 to 110m with an interval of 10m. The purpose of marking the gauges is to facilitate the description in the subsequent result analysis (Section 5). However, we ignored this problem in Section 5. The horizontal distance x was used in Figure 9 to Figure 12 instead of the gauge label. The labels of gauges are used in Figure 9 to Figure 12 in the revised version. The corresponding parts of the article from Line 257 to Line 293 have been carefully revised. Please see the revised version for detail.
Point 6: The term "multiplication coefficient" must be explained in the text. Perhaps, a better word would be "peak pressure factor" or something similar.
Response 6: The problem you pointed out is quite accurate, the word "peak overpressure factor" instead of "multiplication coefficient" is more appropriate in this paper. And there is no necessary explanation for the new concept. It is assumed that the peak overpressure of gauges in Model 1 to Model 3 is Pground, and the peak overpressure of gauges in Model 4 is Pair, then it is defined in this paper "peak overpressure factor" α=Pground/Pair. It reflects the enhancement effect for peak overpressure of different grounds. Necessary explanations have been added to the corresponding places in the revised version. In addition, please see Section 5.4 in the revised version for detail.
Point 7: The phrase "Grounds have the multiplication effect on peak overpressures" does not sound right at all.
Response 7: We are very grateful for your insightful comments and questions, and we are sorry to fail to explain clearly in the original manuscript. The phrase "Grounds have the multiplication effect on peak overpressures" should be replaced by "Grounds have the enhancement effect on peak overpressures".
Point 8: Also, authors might improve paper readability by paying more attention to English (line 63 - Definition, not Definite; line 76 - "which has certain authority" is not understandable; etc.)
Response 8: The word "Definite" in line 63 is replaced by "Definition" in revised version, and the phrase "which has certain authority" in line 76 is replaced by the sentence "This method has certain authority and has been widely used abroad.". Indeed, the English expression in the paper can be better, we would check and polish the whole paper to improve the readability.
Please see the revised version for detail. We hope the revised version can meet your concern.
Sincerely yours,
Yan Wang.

Reviewer 2 Report
Abstract needs to be improved and without non standard parameters such as x and incident angles; Reference for LS-Dyna manual emphasising the version of the code to be provided; Is the modelling approach appropriate, with the pressure calculated as a mean stress; How is the domain size determined; Horizontal distance near line 133 to be changed to radial; Check the units for time; Typo in the material model name, near line 173; Reference for source for figure 6 needs to be provided; Linear rise cannot be referred to as a shock; Change the notation for the units used on the axis from / to []; Check the units for the peak pressure (at the level of MPa) in the context of the shock loading;Author Response
Dear professor:
I am the corresponding author of the manuscript “Investigating different grounds effects on shock wave propagation resulting from near-ground explosion”(ID applsci-565271). We must thank you for the critical comments and constructive recommendations. We feel lucky that our manuscript went to you as the valuable comments from you not only helped us with the improvement of our manuscript, but also raised some thoughtful suggestions.
Based on your comments and suggestions, we have made significant modification on the original manuscript. We have asked for native English speakers to revise the paper before it was submitted this time. We hope the new manuscript will meet your magazine’s standard.
Here are our one-by-one responses to your comments.
Point 1: Abstract needs to be improved and without non standard parameters such as x and incident angles;
Response 1: Thank you very much for your insightful and valuable comments. The abstract has been thoroughly reviewed and revised, and the problems you pointed out have been carefully revised, with the words replaced by appropriate descriptions. Please see the revised version for detail.
Point 2: Reference for LS-Dyna manual emphasising the version of the code to be provided;
Response 2: The version of the code of LS-DYNA is Version 971 reference for LS-DYNA manual.
Point 3: Is the modelling approach appropriate, with the pressure calculated as a mean stress;
Response 3: Sorry for this confusion. In fact, as mentioned in reference 45, the magnitude of peak overpressure of air shock wave is a characteristic parameter to measure the destructive power of liquid rocket explosion accident on target. Therefore, the pressure obtained from the simulation is the peak overpressure rather than the mean stress.
Point 4: How is the domain size determined;
Response 4: Thank you very much for your valuable comment. It is a wonderful question. Mesh size may affect the simulation results to some extent. Theoretically, within a certain range, the smaller the mesh size is, the more accurate the calculation results will be. But the computational amount increases with the decrease of the mesh size. On the other hand, the closer it is to the explosive, the more significant effect of mesh size will be. Therefore, gradient mesh is adopted in this paper. There is no research on how to divide the domain size at present., Reference to the previous experience for modeling, and through a number of trials and comparisons, the air domain is divided into three parts according to the radial, which are 0~30m, 30~70m, 70~129m.
Point 5: Horizontal distance near line 133 to be changed to radial; Check the units for time; Typo in the material model name, near line 173; Reference for source for figure 6 needs to be provided; Linear rise cannot be referred to as a shock; Change the notation for the units used on the axis from / to []; Check the units for the peak pressure (at the level of MPa) in the context of the shock loading;
Response 5: Thank you very much for your valuable comment and we accept it entirely. Horizontal distance has been changed to radial of the whole paper. The units for time are correct after checking. Typo in the material model name near line 173 has been changed in the revised version. Reference for source for Figure 6 has been provided in the revised version. Shock wave is one of the main destructive effects after rocket explosion. The notation for the unit used on the axis from / to [] has been changed. The unit for the peak pressure is MPa.
Please see the revised version for detail. We hope the revised version can meet your concern.
Sincerely yours,
Yan Wang.
